# T and NK cell lymphoma cell lines do not rely on ZAP-70 for survival

Sanjay de Mel[1][�]*, Nurulhuda Mustafa[2][�]*, Viknesvaran Selvarajan[3], Muhammad Irfan Azaman[4], Patrick William Jaynes[4], Shruthi Venguidessane[4], Hoang Mai Phuong[4], Zubaida Talal Alnaseri[4], The Phyu[3,4], Louis-Pierre Girard[5,6], Wee Joo Chng[1,2,4], Joanna Wardyn[4], Ying Li[4], Omer An[4], Henry Yang[4], Siok Bian Ng[3,4], Anand D. Jeyasekharan[1,2,4]

1 Department of Haematology Oncology, National University Cancer Institute Singapore, National University Health System Singapore, Singapore, Singapore, 2 Department of Medicine, Yong Loo Lin School of Medicine, National University of Singapore, Singapore, Singapore, 3 Department of Pathology Yong Loo Lin School of Medicine, National University of Singapore, Singapore, Singapore, 4 Cancer Science Institute of Singapore, National University of Singapore, Singapore, Singapore, 5 School of Medicine, University of Aberdeen, Aberdeen, United Kingdom, 6 Aberdeen Royal Infirmary, NHS Grampian, Scotland, United Kingdom

” These authors contributed equally to this work.
* sanjay_widanalage@nuhs.edu.sg (SM); mdcnm@nus.edu.sg (NM)

**Data Availability Statement:** All relevant data are within the paper and its Supporting Information files.

## Abstract

B-cell receptor (BCR) signalling is critical for the survival of B-cell lymphomas and is a therapeutic target of drugs such as Ibrutinib. However, the role of T-cell receptor (TCR) signalling in the survival of T/Natural Killer (NK) lymphomas is not clear. ZAP-70 (zeta associated protein-70) is a cytoplasmic tyrosine kinase with a critical role in T-cell receptor (TCR) signalling. It has also been shown to play a role in normal NK cell signalling and activation. High ZAP-70 expression has been detected by immunohistochemistry in peripheral T cell lymphoma (PTCL) and NK cell lymphomas (NKTCL). We therefore, studied the role of TCR pathways in mediating the proliferation and survival of these malignancies through ZAP-70 signalling. ZAP-70 protein was highly expressed in T cell lymphoma cell lines (JURKAT and KARPAS-299) and NKTCL cell lines (KHYG-1, HANK-1, NK-YS, SNK-1 and SNK-6), but not in multiple B-cell lymphoma cell lines. siRNA depletion of ZAP-70 suppressed the phosphorylation of ZAP-70 substrates, SLP76, LAT and p38MAPK, but did not affect cell viability or induce apoptosis in these cell lines. Similarly, while stable overexpression of ZAP-70 mediates increased phosphorylation of target substrates in the TCR pathway, it does not promote increased survival or growth of NKTCL cell lines. The epidermal growth factor receptor (EGFR) inhibitor Gefitinib, which has off-target activity against ZAP-70, also did not show any differential cell kill between ZAP-70 overexpressing (OE) or knockdown (KD) cell lines. Whole transcriptome RNA sequencing highlighted that there was very minimal differential gene expression in three different T/NK cell lines induced by ZAP-70 KD. Importantly, ZAP-70 KD did not significantly enrich for any downstream TCR related genes and pathways. Altogether, this suggests that high expression and constitutive signalling of ZAP-70 in T/NK lymphoma is not critical for cell survival or downstream TCR-mediated signalling and gene expression. ZAP-70 therefore may not be a suitable therapeutic target in T/NK cell malignancies.

**Funding:** Dr. Anand Jeyasekharan is a recipient of the Singapore Ministry of Health's National Medical Research Council Transition Award (NMRC/TA/0052/2016). The funders had no role in study design, data collection and analysis, decision to publish, or preparation of the manuscript.

**Competing interests:** the authors have declared that no competing interests exist.

## Introduction

B-cell receptor (BCR) signalling is critical for the survival of B-cell malignancies and is a therapeutic target for drugs such as ibrutinib [1–3]. In contrast, the role of T-cell receptor (TCR) signalling in the survival of peripheral T cell and NK cell lymphomas (PTCL and NKTL) remains unclear. The normal function of the TCR complex is to transduce signals promoting cellular survival and proliferation in response to external stimuli [4, 5]. A key molecule within this ensemble is the TCR zeta chain-associated protein of 70 kDa (ZAP-70) [6].

ZAP-70 is a cytoplasmic tyrosine kinase expressed primarily expressed in T and NK cells [7, 8]. Upon recruitment to the TCR and subsequent phosphorylation, active ZAP-70 itself phosphorylates tyrosines on two scaffold proteins: cytoplasmic SH2 domain-containing leukocyte protein of 76 kDa (SLP-76) and the transmembrane linker for the activation of T cells (LAT) protein [9, 10]. ZAP-70 mediated phosphorylation of LAT and SLP-76 facilitates the recruitment and assembly of the TCR signalosome, and subsequent T cell activation involving several downstream partners including p38 mitogen activated protein kinase (MAPK) [6, 11]. Though NK cells do not express a single dominant—and defining—antigen receptor, some of their receptors do bear immunoreceptor tyrosine based activation motifs (ITAM) and rely on similar molecular machinery as TCR signalling, including ZAP-70 dependant phosphorylation [12].

In both mice and humans, loss of ZAP-70 results in a severe combined immune deficiency defined by loss of functioning peripheral T cells [13–17]. In murine NK cells, the absence of both ZAP-70 and Syk does not prevent overall cellular development or lytic activity but does impair functions mediated by ITAM-coupled receptors [18]. These data highlight the importance of ZAP-70 for normal T cell development.

Despite being absent in normal B cells [7] ZAP-70 functions as an adaptor protein promoting BCR signalling in a subset of chronic lymphocytic leukaemia (CLL), where its presence confers inferior clinical outcomes [19–21]. Furthermore, Dielschneider et al. observed that inhibition of ZAP-70 via the use of gefitinib was cytotoxic to CLL cells [22]. There is also emerging evidence that TCR signalling plays a role in the pathogenesis of T and NK cell malignancies [23, 24], however, the role of ZAP-70 per se in these tumours remains unclear. We therefore examined the role of ZAP-70 in TCR signalling for survival of PTCL/NKTL cell lines.

## Materials and methods

### Cell culture

T cell lymphoma (TCL) cell lines (JURKAT, KARPAS-299) and NKTL cell lines (KHYG-1, HANK-1, NK-YS, SNK-1, and SNK-6) were used in this study. KHYG-1 cells were obtained from the JCRB cell bank (JCRB0156); HANK-1 cells were a kind gift from Dr Yoshitoyo Kagami of Aichi Cancer Centre Hospital, Nagoya, Japan; NK-YS cells were a kind gift from Dr Yok-Lam Kwong of Queen Mary Hospital, Hong Kong; SNK-1 and SNK-6 cells were a kind gift from Dr Norio Shimizu of Tokyo Medical and Dental University, Japan. JURKAT, KARPAS-299, KHYG-1 and NK-YS were propagated in RPMI-1640 medium (Thermo Scientific, USA), supplemented with 10% foetal bovine serum (PAA Labs, Austria). HANK-1, SNK-1 and SNK-6 cell lines were maintained in Artemis-2 medium (Nihon Techno Service Co. Ltd, Japan) supplemented with 2% human serum (PAA Labs, Austria). KHYG-1, HANK-1 and NK-YS were maintained with 100 U/ml of recombinant human IL-2 while SNK-1 and SNK-6 were supplemented with 700 U/ml of recombinant human IL-2 (Miltenyi Biotec, Germany). Hela (cervical cancer), THP-1 (monocytic leukaemia) and RAJI (Burkitt lymphoma) cell lines were used as controls for cell lines that are negative for ZAP70. All cell lines were incubated at 37˚C in a humidified atmosphere of 5% $CO_2$.

## RNA extraction and real-time quantitative PCR analysis

Total RNA was prepared using the miRNeasy Mini Kit (Qiagen GmbH, Germany) protocol with the DNaseI treatment included. A reverse transcription reaction was carried out by using the Maxima First Strand cDNA Synthesis Kit (Thermo Scientific, USA). Real-time fluorescence monitoring of the PCR products was assayed with Power SYBR Green PCR Master Mix (Thermo Scientific, USA) and gene-specific primers (*ZAP-70*, *AHNAK2*, *PMEPA1*, *KLHD7CB*, *PRDX4*, *CMYC*, *ATF5*, *GAPDH* and *β-Actin*). The assay was performed using the 7500 Fast Real-Time PCR (RT-PCR) System (Thermo Scientific, USA). Gene expression levels were calculated using $2^{(-\Delta\Delta CT)}$ by comparing the amount of endogenous *GAPDH and β-Actin* in the same sample.

## Western blot analysis

Treated cell pellets were suspended in lysis buffer with a cocktail of protease inhibitors (Promega, USA). Protein detection by western blot was performed by electrophoretic transfer of equal amounts of SDS-PAGE separated proteins to polyvinyl difluoride (PVDF) membranes (Biorad, USA), incubated with the following primary antibodies: ZAP-70 #2705, LAT #4553, Phosphorylated-LAT #3584, SLP76 #4958 Phospho-SLP76 #92711, p-ZAP70 #2717, p38 MAPK #9212 (Cell Signalling Technology, USA) Actin (C-2; Santa Cruz Biotechnology Inc., USA) and GADPH (6C5; Santa Cruz Biotechnology Inc., USA). Either a horseradish peroxidase (HRP)-conjugated anti-rabbit secondary antibody (Santa Cruz Biotechnology Inc., USA) or an HRP-conjugated mouse IgG kappa binding protein (m-IgGκ BP) (Santa Cruz Biotechnology Inc., USA) was used for primary antibody detection via chemiluminescence (GE Healthcare, UK).

## ZAP-70 knockdown

siRNA transfection was performed using Neon® Transfection System (Thermo Scientific, USA) following the manufacturer's instructions. PTCL and NKTL cell lines were electroporated with 50nM of non-targeting siRNA control and siZAP-70 (Dharmacon Inc., USA) using the optimized pulse conditions.

## ZAP-70 overexpression

The overexpression construct pcDNA-ZAP-70 was a kind gift from Prof Dimitar Efremov from The international Centre for Genetic Engineering & Biotechnology (ICGEB) in Italy. pcDNA-ZAP-70 transfection was performed using Neon® Transfection System (Thermo Scientific, USA) following the manufacturer's instructions. NKYS cells (approximately $2 \times 10^6$ cells) were electroporated with 5μg of pcDNA ZAP-70. Cells were cultured in a 6-well plate. After 48 hours, the transfected cells were exposed to 1 mg/ml of G-418. After 4 days, 70% of the cell suspension volume was removed and replaced with media containing G-418. Cells were maintained under G-418 selection for at least two weeks before a sample of cells were collected to verify ZAP-70 overexpression via Western blot. The cell population was subsequently maintained in culture under 1mg/ml of G-418 selection.

## Flow cytometric analysis for apoptosis

The apoptotic cell death analyses were carried out using Annexin-V-APC and propidium iodide (PI) detection systems. The staining of apoptotic cells was assayed using Annexin-V Apoptosis Detection Kit (BD Pharmigen, USA) according to manufacturer's instructions and

the analysis was performed on a BD LSR II (Becton Dickinson, USA) flow cytometer, using BD FACSDiva™ software.

### Cell viability assays

For the cell viability assays, ZAP-70-null cells (THP-1, RAJI, SCI and OCI-LY19), T cells (JUR-KAT and KARPAS-299) and NKTL cells (KHYG-1, HANK-1, NK-YS, SNK-1 and SNK-6) were plated in 96-well plates and treated with increasing concentrations of Gefitinib. At 72-hours, cells were lysed with CellTiter-Glo® 2.0 Assay reagent (Promega Corp., USA) and luminescence was read using a plate reader (TECAN Infinite 200 Pro, Switzerland). Cell growth percentage was calculated relative to DMSO treated cells. For cell viability assays using ZAP-70 overexpressing cells, both wild type (WT) and ZAP-70 overexpressing (OE) NKYS cells were plated into 96 well plates and treated with increasing concentrations of Etoposide. After 72 hours, viability was assessed with CellTiter-Blue® (Promega Corp, USA) and read using a plate reader (TECAN Infinite 200 Pro, Switzerland). Cell growth percentage was calculated relative to DMSO treated cells. Prism (GraphPad Software, USA) was used to calculate and plot the IC50 for each biological replicate of Etoposide treated WT and ZAP-70 OE NYKS cells.

### RNA-sequencing

RNA was prepared from knockdown (KD) and WT NKTCL cell lines. The analysis was performed using DEseq2, to assume a hit is significant the p-value has to be above 0.05 and reach at least log2 fold change = 2, so the p-value -log10 of 0.05.

## Results

### ZAP-70 is expressed in TCL and NKTL cell lines but not in cell lines of B-cell malignancies

Upon assessment of the protein levels of ZAP-70, we found that all cell lines belonging to the TCL and NKTL groups expressed ZAP-70. ZAP-70 was absent from the various control cell lines, validating their use as such (**Fig 1A**). The assessment of cells lines derived from B-cell malignancies, using JURKAT cells as a positive control for ZAP-70 presence, showed that they showed no expression of ZAP-70 (**Fig 1B**). Consistent with our findings in Fig 1A and 1B, ZAP-70 mRNA transcripts were expressed in our TCL and NKTL cell lines but not in the control cell lines (**Fig 1C**). Of note, comparing between cell lines, the ZAP-70 mRNA expression pattern was comparable to the protein expression pattern. Karpas 299 is derived from Anaplastic large cell lymphoma with a NPM1-ALK fusion. Our observation that Karpas 299 expresses ZAP-70 is consistent with a previous report [25].

### Knockdown of ZAP-70 suppresses phosphorylation of direct substrates SLP-76, p38 MAPK and LAT

We selected a high ZAP-70 expressing NKTL cell line, KHYG and evaluated the effect of silencing ZAP-70 protein in NKTL cell lines. Immunoblotting was performed to determine the expression levels of phosphorylated and total ZAP-70, as well as ZAP-70 target substrates within the TCR signalling pathway (phosphorylated and total SLP-76, LAT and p38 MAPK). We demonstrated that ZAP-70 is constitutively phosphorylated and active in KHYG resulting in downstream phosphorylation of its substrates. Depletion of ZAP-70 protein by siRNA at 48 hours showed a clear reduction in the phosphorylation of SLP-76 and p38 MAPK but not LAT (**Fig 2A**). The suppression of LAT phosphorylation due to ZAP-70 knockdown is only evident

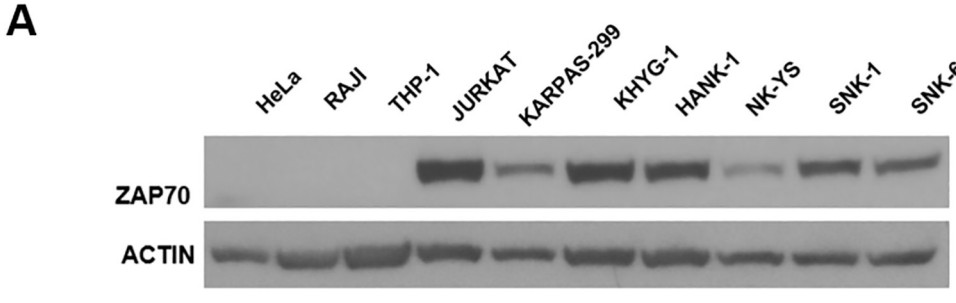

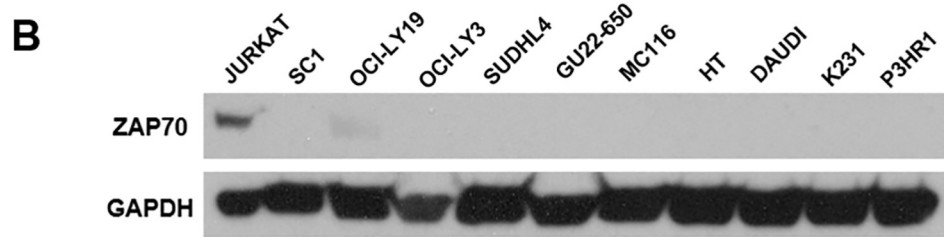

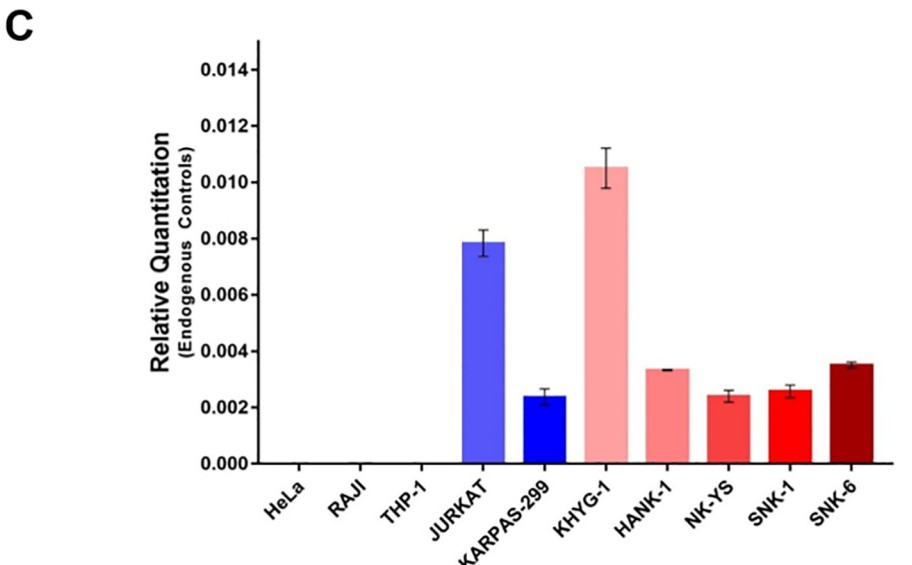

**Fig 1. ZAP70 is expressed in cell lines derived from TCL and NKTL cells but not in cell lines derived from B-cell malignancies.** (A) ZAP70 protein expression as assessed by Western blotting in Hela, RAJI, THP-1, T cell lines (JURKAT, KARPAS-299) and NKTL cell lines (KHYG-1, HANK-1, NK-YS, SNK-1, SNK-6). β–ACTIN was used as loading control. (B) ZAP70 protein expression as assessed by Western blotting in JURKAT, SC1, OCI-LY19, OCI-LY3, SUDHL4, GU22-650, MC116, HT, DAUDI, K231, P3HR1. GAPDH was used as loading control. (C) ZAP70 mRNA was measured by real time RT-PCR using gene-specific primers (ZAP70, GAPDH and β-ACTIN) in Hela, RAJI, THP-1 and JURKAT, KARPAS-299 and KHYG-1, HANK-1, NK-YS, SNK-1,SNK-6 cell lines. GAPDH and β–ACTIN were used as housekeeping genes.

at 72h. Interestingly, at 72h following ZAP-70 knockdown, we also observed a restoration of the phosphorylation of SLP-76 and p38 MAPK protein expression despite the continued depletion of ZAP-70 by the siRNA knockdown (Fig 2A). Raw western blots are presented under (S1 Raw images).

## Knockdown of ZAP-70 does not affect survival of TCL or NKTL cell lines

Once we had determined that the knockdown of ZAP-70 protein can suppress phosphorylation of target components in the TCR signalling pathway in NKTL cell lines, we proceeded to examine the effect of ZAP-70 KD on cell survival. We first studied the effect of ZAP-70 KD on apoptosis using Annexin V and propidium iodide (PI) staining in the KHYG cell line. We observed that the percentage of apoptotic cells (defined as Annexin V+/PI- or Annexin V+/PI +) was not markedly different between control and ZAP-70 knockdown conditions. Furthermore, a time-dependent knockdown of ZAP-70 confirmed no further induction of apoptosis (Fig 2B). The positive control treatment with 5μM of Obatoclax, a BH3 mimetic demonstrated that the cells can successfully undergo apoptosis.

To validate that this observation is not cell line dependent, we performed ZAP-70 knockdown in a larger range of T and NK lymphoma cell lines. The efficacy of the ZAP-70 protein knockdown was demonstrated with two concentrations of siRNA (25nM and 50nM). We confirm that both concentrations demonstrate similar efficacy in disrupting ZAP-70 expression at both protein (Fig 2C) and mRNA (Fig 2D) levels. Subsequently, the effect of the depletion of ZAP-70 protein expression in this wide range of cell lines was analysed by Annexin V/PI staining. In line with the earlier observation, depletion of ZAP-70 in all the cell lines tested did not result in any induction of apoptosis (Fig 2E).

Apoptosis is a cell death assay and cell survival and proliferation may potentially be inhibited by other mechanisms such as cell cycle arrest. To confirm that the depletion of ZAP-70 protein had no other effects on overall cell survival, we also studied the effect of ZAP-70 knockdown over the time course of 48h and 72h via a cell viability and growth assay. Once again, we confirmed that there was no reduction in cell viability where ZAP-70 had been depleted as compared to controls (Fig 2F).

## RNA sequencing demonstrates that depletion of ZAP-70 protein induces minimal gene expression change and does not significantly regulate the expression of genes associated with downstream TCR signalling pathways in T/NK Lymphoma

Given that ZAP-70 KD did not affect the survival of NKTL cell lines, we performed RNA sequencing on three NKTL cell lines to elucidate the functional role of ZAP-70. The effect of ZAP-70 depletion on TCR-associated downstream pathways and genes in NKTL was of particular interest. Firstly, we first performed a Gene Set Enrichment Analysis (GSEA) utilising the curated gene sets (C2) from Molecular Signatures Database (Broad Institute). We found that among the three NKTL cell lines there were no overlapping gene sets that were significantly enriched upon ZAP-70 knockdown (Table 1, Fig 3A). In fact, only HANK-1, demonstrated a significant negative enrichment for 178/4435 gene sets while both NKYS and SNK-1, showed no significantly enriched gene sets. Importantly, all three NKTL cell lines demonstrated that ZAP-70 knockdown did not result in any significant enrichment for TCR related downstream pathways (Table 1, Fig 3A).

Subsequently, we performed gene set enrichment analysis of the RNA-sequencing data with the Hallmark gene sets which represent specific and well-defined biological states or processes (Table 2, Fig 3B).

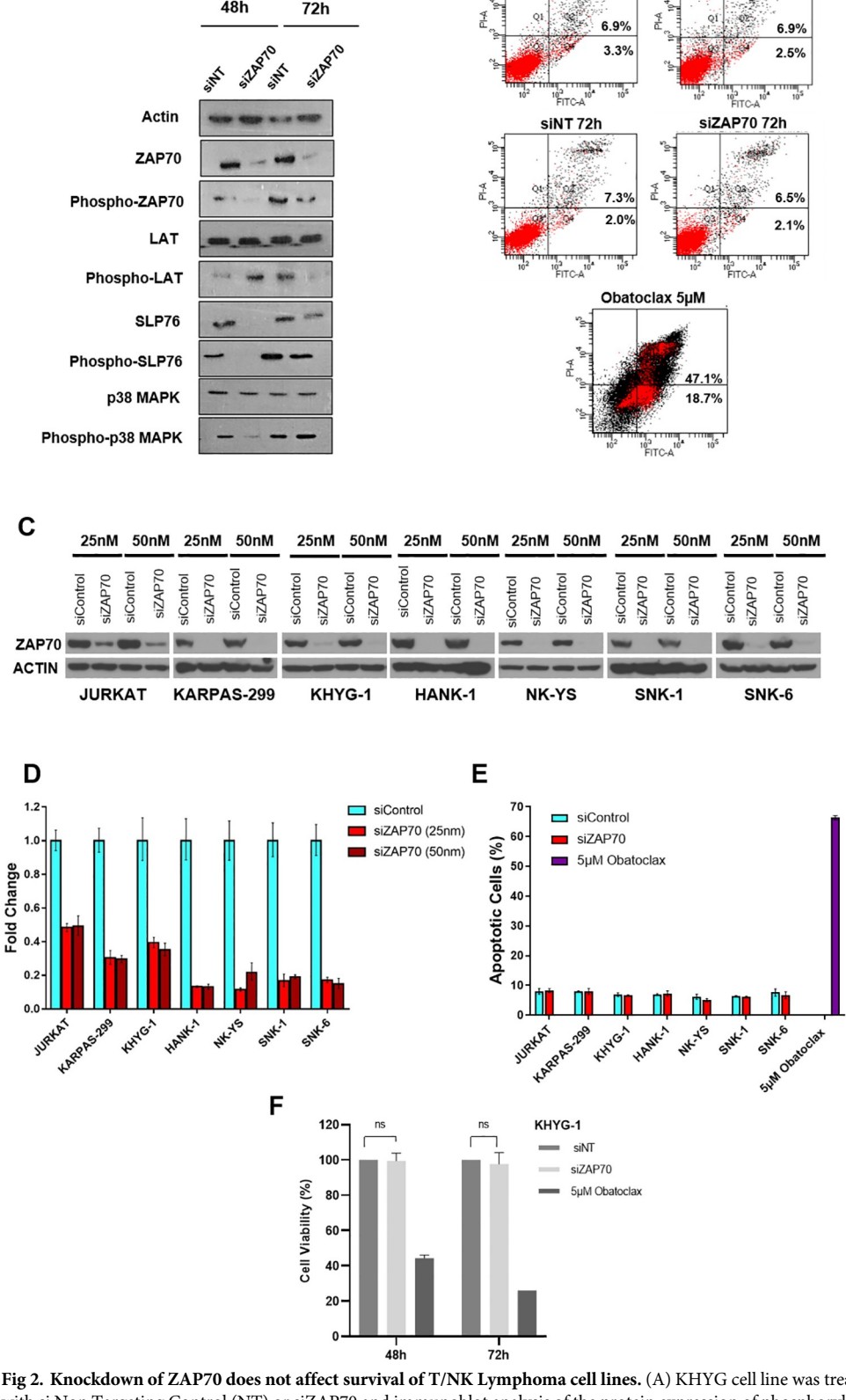

**Fig 2. Knockdown of ZAP70 does not affect survival of T/NK Lymphoma cell lines.** (A) KHYG cell line was treated with si Non Targeting Control (NT) or siZAP70 and immunoblot analysis of the protein expression of phosphorylated

ZAP-70, total Zap-70 phosphorylated LAT, total LAT, phosphorylated SLP-76, total SLP-76, phosphorylated p38 MAPk, total p38MAPK were evaluated with Actin as the loading control. (B) KHYG cells which have been either treated with the siNT or si-ZAP-70 was evaluated for apoptosis by Annexin V PI staining. Obatoclax is a positive control demonstrating induction of apoptosis. (C) ZAP70 siRNA knockdown was performed in a wider range of cell lines. Protein expression in WT and ZAP70 knockdown TCL cells (JURKAT and KARPAS-299) and NKTL cells (KHYG1, HANK-1, NK-YS, SNK-1 and SNK-6) cell lines were analysed by immunoblotting. β– ACTIN was used as a loading control. (D) ZAP70 mRNA was measured in control and ZAP70 siRNA conditions by RT-PCR in TCL cells (JURKAT and KARPAS-299) and NKTL cells (KHYG1, HANK-1, NK-YS, SNK-1 and SNK-6) (E) TCL cells (JURKAT and KARPAS-299) and NKTL cells (KHYG1, HANK-1, NK-YS, SNK-1 and SNK-6) were stained with Annexin V-APC and PI and analyzed by flow cytometry. Obatoclax 5μM treatment is a positive control demonstrating induction of apoptosis. (F) KHYG1cells which have been either treated with the siNT or si-ZAP-70 was evaluated by Cell-Titer Glo (Promega) after 48h.

To further understand these findings, we interrogated genes that were differentially regulated by ZAP-70 knockdown in each of these three NKTL cell lines. These genes have been represented in a volcano plot. Except for the ZAP-70 gene which was robustly and consistently downregulated in all three NKTL cell lines, ZAP-70 knockdown resulted in minimal changes in gene expression (**Fig 3C**). qRT-PCR was performed to validate the results of the RNA sequencing and confirmed that the genes AHNAK2 and PMEPA1 were correspondingly downregulated and KLHDC7B upregulated when ZAP-70 was depleted by siRNA (**Fig 3D**).

## Overexpression of ZAP-70 does not promote survival of NKTL cell lines or increase their sensitivity to etoposide

The loss of Myc targets, G2/M activity and E2F targets (Table 2) (suggesting cell cycle arrest) is consistent with a recent report that NK cells downregulate ZAP-70 in response to DNA damage [26]. We postulated that downregulation of ZAP-70 is a protective mechanism and hypothesised that forced overexpression of ZAP-70 with concomitant activation of DNA damage might promote cell death in NKTL. We selected the NKYS cell line to pursue this investigation due to its relatively low expression of ZAP-70 allowing for forced overexpression to be within a physiological range.

Immunoblot analysis confirmed that the overexpression of ZAP-70 protein corresponds with an increase in the phosphorylation of ZAP-70 protein as well as phosphorylation of ZAP-70 target substrates in the TCR signalling pathway, SLP-76 and p38 MAPK (**Fig 4A**). Overexpression of ZAP-70 also induced an upregulation of SLP-76 and LAT total protein expression. Additional stimulation of the TCR pathway with OKT3 over 4h and 18h in these cells corroborates with an increase in phospho-ZAP-70 expression as well as an increase of phospho-LAT and phospho- SLP-76 thereby confirming that ZAP-70 functionally mediates the upstream TCR signalling pathways in NKTL (**S1 Fig**). We studied the growth curve of the empty vector (EV) and ZAP-70 overexpressing stable cell lines and found that there was no growth advantage conferred by ZAP-70 overexpression despite the enhanced signalling (**Fig 4B**).

Having confirmed the effects of ZAP-70 OE, we subsequently tested the hypothesis that upregulation of ZAP-70 protein may sensitise NKTL cells to DNA damage inducing agents such as etoposide. Thus, we performed an etoposide dose-response cell viability assay to compare the response to etoposide between wild type and ZAP-70 OE cell lines. We found no significant difference in the IC50s between the two cell lines (**Fig 4C**) suggesting that ZAP-70 may not influence the response to etoposide in NKTL. To gain further insight regarding the potential role of E2F and MYC gene targets in ZAP-70 signalling, we selected a few top target genes, PRDX4, CMYC and ATF5 from the gene sets that were identified to be negatively enriched in the GSEA Hallmark analysis and performed qRT-PCT to validate the expression

**Table 1. Gene set enrichment analysis of siZAP-70 vs siNT in HANK-1, NKYS and SNK-1 NKTL cell lines [C2 gene set analysis].**

| HANK-1 | | | | |
|---|---|---|---|---|
| 0/4435 gene sets are positively enriched at FDR < 10% | | | | |
| 178/4435 gene sets are negatively enriched at FDR < 10% | | | | |
| GENE SET | SIZE | NES | NOM p-val | FDR q-val |
| PID_CD8_TCR_DOWNSTREAM_PATHWAY | 47 | 1.663 | 0.004 | 0.262 |
| PID_TCR_CALCIUM_PATHWAY | 22 | 1.418 | 0.069 | 0.561 |
| PID_TCR_RAS_PATHWAY | 14 | 1.169 | 0.264 | 0.853 |
| PID_TCR_JNK_PATHWAY | 14 | 0.603 | 0.936 | 1.000 |
| PID_TCR_PATHWAY | 61 | 0.547 | 1.000 | 1.000 |
| BIOCARTA_TCR_PATHWAY | 41 | -0.793 | 0.786 | 0.999 |
| PID_CD8_TCR_PATHWAY | 48 | -0.586 | 0.991 | 1.000 |
| REACTOME_TCR_SIGNALING | 112 | -0.491 | 1.000 | 1.000 |
| SNK-1 | | | | |
| 0/4534 gene sets are positively enriched at FDR < 10% | | | | |
| 0/4534 gene sets are negatively enriched at FDR < 10% | | | | |
| GENE SET | SIZE | NES | NOM p-val | FDR q-val |
| PID_TCR_CALCIUM_PATHWAY | 25 | 1.546 | 0.024 | 0.828 |
| PID_TCR_RAS_PATHWAY | 14 | 1.209 | 0.260 | 0.916 |
| PID_TCR_JNK_PATHWAY | 14 | 0.553 | 0.955 | 1.000 |
| PID_CD8_TCR_PATHWAY | 48 | -1.436 | 0.044 | 1.000 |
| PID_TCR_PATHWAY | 60 | -1.389 | 0.065 | 1.000 |
| BIOCARTA_TCR_PATHWAY | 40 | -1.175 | 0.225 | 1.000 |
| PID_CD8_TCR_DOWNSTREAM_PATHWAY | 47 | -0.951 | 0.535 | 1.000 |
| REACTOME_TCR_SIGNALING | 109 | -0.881 | 0.718 | 1.000 |
| NKYS | | | | |
| 0/4481 gene sets are positively enriched at FDR < 10% | | | | |
| 0/4481 gene sets are negatively enriched at FDR < 10% | | | | |
| GENE SET | SIZE | NES | NOM p-val | FDR q-val |
| PID_TCR_CALCIUM_PATHWAY | 21 | 0.746 | 0.831 | 1.000 |
| PID_CD8_TCR_DOWNSTREAM_PATHWAY | 45 | 0.726 | 0.890 | 1.000 |
| PID_TCR_RAS_PATHWAY | 14 | 0.680 | 0.850 | 1.000 |
| PID_TCR_JNK_PATHWAY | 13 | 0.420 | 0.992 | 1.000 |
| BIOCARTA_TCRA_PATHWAY | 11 | -1.061 | 0.387 | 1.000 |
| REACTOME_TCR_SIGNALING | 113 | -1.057 | 0.320 | 1.000 |
| BIOCARTA_TCR_PATHWAY | 42 | -0.848 | 0.714 | 1.000 |
| PID_CD8_TCR_PATHWAY | 49 | -0.719 | 0.919 | 1.000 |
| PID_TCR_PATHWAY | 63 | -0.540 | 1.000 | 1.000 |

TCR related pathways are not regulated by ZAP-70 knockdown.

levels of these targets. Our results show that there is no significant downregulation of PRDX4 and MYC induced by ZAP-70 knockdown while ATF5 was just slightly upregulated. (**S2A Fig**). Conversely, overexpression of ZAP70 confers either no significant change or slight downregulation of these genes contrary to what may be expected (**S2B Fig**). However, it is also worth noting that these mRNA expression changes are less than 1.5 fold and may not have a significant impact on the survival and death signalling pathways as observed by our functional cell viability and apoptosis assays.

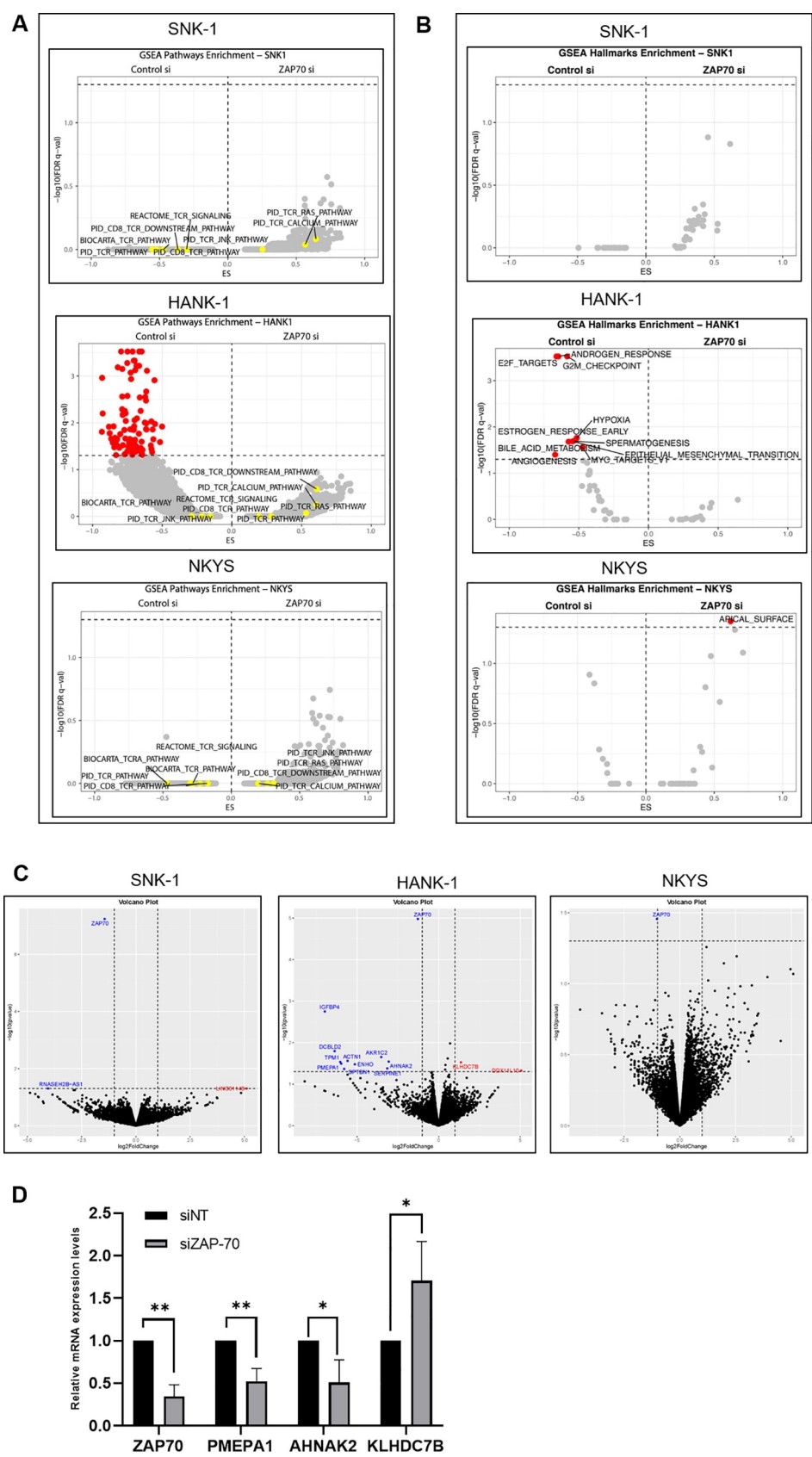

**Fig 3. RNAseq highlights that there are not many genes differentially regulated with the depletion of ZAP-70 protein.** (A) GSEA plots showing the analysis plot of genes sets significantly enriched in the NKTL cell lines (HANK-1, NK-YS, SNK-1) after ZAP-70 knockdown via siRNA utilising the (A) C2 (curated gene sets) (B) Hallmark gene set parameters for gene set analysis. (C) Differential gene expression analysis is represented by individual volcano plots for indicated NKTL cell lines shows that there are very few genes that are differentially expressed between the si-non targeting and si-ZAP70 in the three cell lines. (D) KHYG cells were treated with siNT or siZAP70 and then the effect on the mRNA expression levels of ZAP70, PMEPA1, AHNAK2 and KLHD7CB measured via qRT-PCR.

## Gefitinib does not preferentially induce cell death in T/NK lymphoma cell lines overexpressing ZAP-70

Gefitinib has been shown to selectively induce cell death in ZAP-70 expressing CLL [22]. In Fig 1 we have shown a range of T and NK lymphomas with varying levels of ZAP-70

**Table 2. Gene set enrichment analysis of siZAP-70 vs siNT in HANK-1, NKYS and SNK-1 NKTL cell lines [Hallmark gene set analysis].**

| HANK1 | | | | |
|---|---|---|---|---|
| 0/50 gene sets are positively enriched at FDR < 25% | | | | |
| 21/50 gene sets are negatively enriched at FDR < 25% | | | | |
| **GENE SET** | **SIZE** | **NES** | **NOM p-val** | **FDR q-val** |
| **HALLMARK_E2F_TARGETS** | **199** | **-2.246** | **0.000** | **0.000** |
| **HALLMARK_G2M_CHECKPOINT** | **195** | **-1.955** | **0.000** | **0.000** |
| HALLMARK_ANDROGEN_RESPONSE | 86 | -1.931 | 0.000 | 0.000 |
| HALLMARK_EPITHELIAL_MESENCHYMAL_TRANSITION | 114 | -1.705 | 0.000 | 0.020 |
| HALLMARK_BILE_ACID_METABOLISM | 80 | -1.680 | 0.000 | 0.021 |
| HALLMARK_HYPOXIA | 145 | -1.675 | 0.000 | 0.018 |
| HALLMARK_ESTROGEN_RESPONSE_EARLY | 143 | -1.662 | 0.000 | 0.017 |
| HALLMARK_SPERMATOGENESIS | 80 | -1.624 | 0.005 | 0.020 |
| **HALLMARK_MYC_TARGETS_V1** | **198** | **-1.590** | **0.005** | **0.027** |
| SNK1 | | | | |
| 2/50 gene sets are positively enriched at FDR < 25% | | | | |
| 0/50 gene sets are negatively enriched at FDR < 25% | | | | |
| **GENE SET** | **SIZE** | **NES** | **NOM p-val** | **FDR q-val** |
| HALLMARK_HEDGEHOG_SIGNALING | 31 | 1.603 | 0.020 | 0.148 |
| HALLMARK_MYC_TARGETS_V1 | 198 | 1.541 | 0.000 | 0.131 |
| *HALLMARK_G2M_CHECKPOINT* | *196* | *-0.488* | *1.000* | *1.000* |
| *HALLMARK_E2F_TARGETS* | *200* | *-0.458* | *1.000* | *0.999* |
| NKYS | | | | |
| 6/50 gene sets are positively enriched at FDR < 25% | | | | |
| 2/50 gene sets are negatively enriched at FDR < 25% | | | | |
| **GENE SET** | **SIZE** | **NES** | **NOM p-val** | **FDR q-val** |
| HALLMARK_PANCREAS_BETA_CELLS | 18 | 1.637 | 0.007 | 0.081 |
| HALLMARK_APICAL_SURFACE | 31 | 1.627 | 0.008 | 0.045 |
| HALLMARK_HEDGEHOG_SIGNALING | 25 | 1.590 | 0.017 | 0.053 |
| HALLMARK_KRAS_SIGNALING_DN | 113 | 1.510 | 0.003 | 0.087 |
| HALLMARK_MYOGENESIS | 151 | 1.432 | 0.011 | 0.158 |
| HALLMARK_WNT_BETA_CATENIN_SIGNALING | 29 | 1.386 | 0.077 | 0.209 |
| HALLMARK_EPITHELIAL_MESENCHYMAL_TRANSITION | 128 | 1.268 | 0.090 | 0.493 |
| **HALLMARK_E2F_TARGETS** | **200** | **-1.562** | **0.000** | **0.124** |
| **HALLMARK_MYC_TARGETS_V1** | **197** | **-1.442** | **0.005** | **0.146** |

E2F and Myc pathways are negatively enriched in ZAP-70 knockdown cell lines.

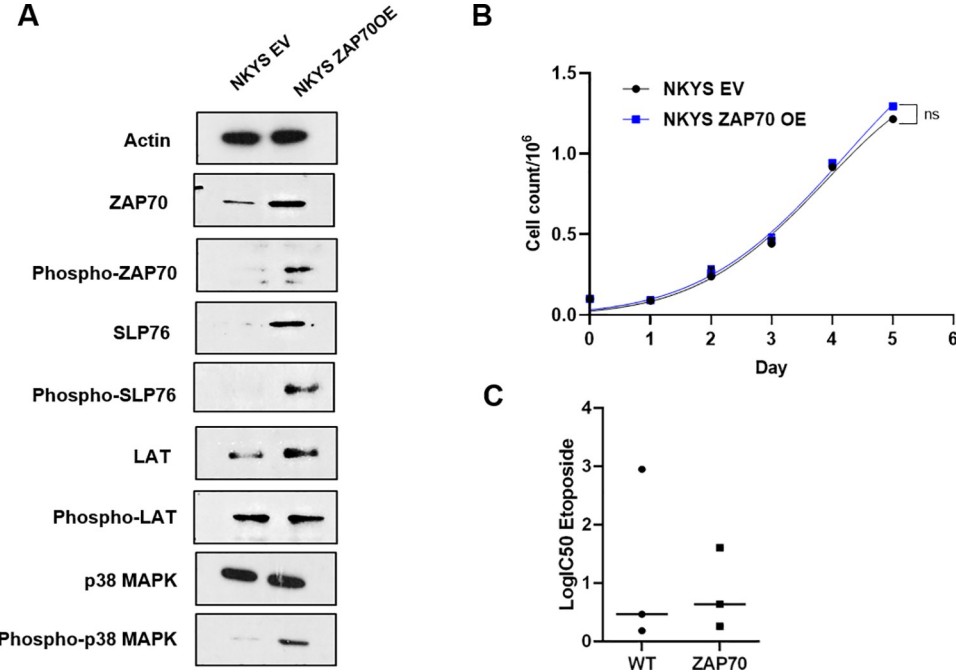

**Fig 4. Overexpression of ZAP70 does not alter the sensitivity of NKT cells to etoposide or increase cell survival of NK/T lymphoma cells.** (A) NKYS cells were stably transfected with EV or ZAP-70. Activation of the TCR signalling components were analysed by Immunoblotting performed on these cell lines and the following proteins detected— phosphorylated and total ZAP70 as well as target substrates of ZAP-70, phosphorylated and total LAT, phosphorylated and total SLP76, phosphorylated and total p38 MAPK (B) The total number of live cells were determined by tryphan blue exclusion for EV and ZAP-70 overexpressing cells for the days indicated. (C) Comparison of IC50 values to etoposide treatment for 24 hours between wild type (WT) NKYS and NKYS with ZAP70 overexpression (ZAP70). Viability was assessed by Cell Titre blue.

expression. We therefore evaluated the effect of Gefitinib on TCL and NKTL by performing a gefitinib dose-response viability assay on our cell lines to investigate whether this demonstrated a preferential targeting effect on high-expressing ZAP-70 cell lines. We did not observe significant sensitivity to Gefitinib in cell lines with high ZAP-70 expression (**Fig 5A**). To confirm this observation, stably transfected NKYS EV and ZAP-70 OE cell lines were also treated with increasing doses of Gefitinib. There was no differential effect of Gefitinib on cell viability of the EV or ZAP-70 OE cell lines (**Fig 5B**), while the same concentration was able to inhibit survival of Jurkat cells by almost 50% (**Fig 5C**). Conversely, ZAP-70 KD in KHYG and HANK-1 did not confer further resistance to Gefitinib treatment (**S3 Fig**).

## Discussion

We demonstrate for the first time that T and NK cell lymphoma cell lines are not reliant on ZAP-70 for survival. It has been hypothesized that PTCL are reliant on TCR signalling for survival based on their ubiquitous expression of TCR signalling proteins [27]. Pre-clinical studies have shown that PTCL are sensitive to inhibition of SYK [28] phosphoinositide -3 kinase [29] and protein kinase C [30] which signal downstream of ZAP-70. There have been no studies specifically investigating the role of ZAP-70 as a driver of PTCL or NKTL.

Our finding of differential ZAP-70 expression between T/NK and B cell lymphoma cell lines is consistent with our current understanding of the biology of B and T-cell signalling [1, 6]. The fact that ZAP-70 depletion did not affect cell viability or proliferation of T/NK cell lymphoma cell lines is however unexpected. ZAP-70 KD resulted in transient reductions in

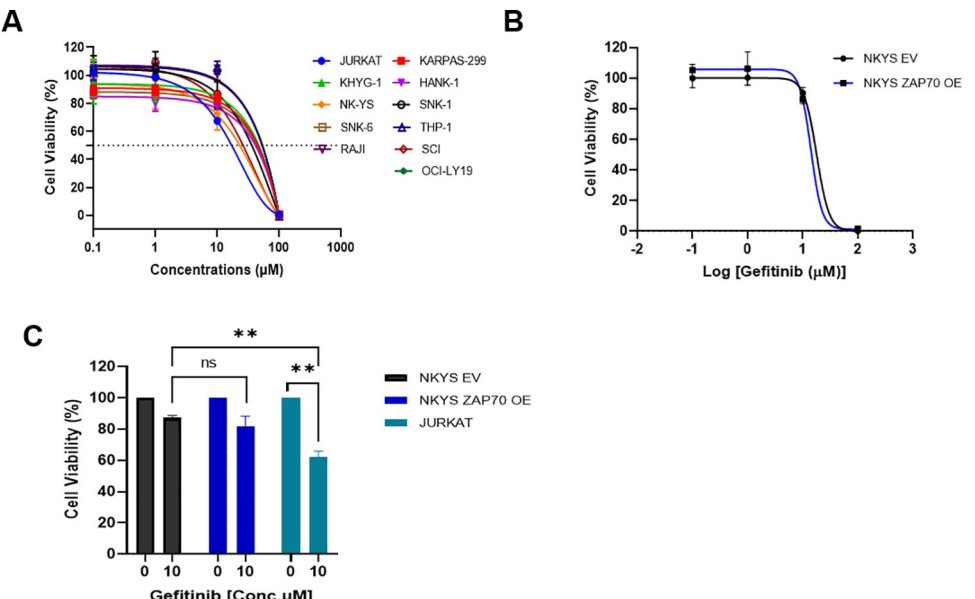

**Fig 5. Gefitinib does not preferentially target ZAP-70 overexpressing NKTL.** (A) Gefitinib dose response curve for TCL cell lines (JURKAT, KARPAS-299), NKTL cell lines (KHYG1, HANK-1, NK-YS, SNK-1, SNK-6), monocytic leukemic cell line (THP-1) and B cell malignant cell lines (RAJI, SCI, OCI-LY19) (B) NKYS cell lines which have been stably transfected with EV or ZAP70 were treated with a range of Gefitinib concentrations and cell viability assessed by Cell–Titer Glo (Promega) (C) Jurkat a high ZAP-70 expressing T cell leukemia cell line is a positive control with growth inhibition induced by Gefitinib.

phosphorylation of some ZAP-70 substrates whose expression was restored at 72 hours despite the continued depletion of the ZAP-70 protein. This suggests that the cells were able to activate TCR signalling independently of ZAP-70 and that the loss of ZAP-70 mediated signalling in the TCR pathway may be compensated by the cell through other proteins or pathways.

Although ZAP-70 is widely accepted to be essential for normal TCR signalling, some subsets of normal T-cells may be less reliant on ZAP-70 than others [31]. It has been suggested that the SYK kinase can compensate for ZAP-70 deficiency to a certain extent in some subsets of non-malignant T-cells [31]. Further studies are required to determine if a similar phenomenon occurs in T/NK Lymphoma. A recurrent chromosomal translocation resulting in the ITK-SYK [t(5;9)(q33;q22)] fusion transcript is commonly observed in PTCL [32]. This has been shown to produce a catalytically active fusion protein that phosphorylates SLP-76, LAT, and other key molecules in the TCR signalosome [32–34]. This fusion protein is able to propagate signals that mimic those of the TCR, without recourse to ZAP-70 and other normal activation processes. Transgenic expression of this protein in mice resulted in a phenotype similar to human T cell lymphoma [33, 35, 36].

Shan et al described ZAP-70 independent TCR signalling in the ZAP-70 negative cell line P116 [37]. TCR stimulation by anti -CD3 antibodies resulted in activation of Erk, Raf-1 and MEK-1 in the absence of LAT phosphorylation in P116 cells. ZAP-70 independent TCR signalling in P116 cells was shown to be dependent on protein kinase C. These findings suggest that some T cell malignancies can survive in the absence of ZAP-70 expression, and canonical TCR-mediated signalling. The effect of gefitinib on ZAP-70 expressing CLL cells may be a phenomenon specific to CLL and not applicable to PTCL. It may also be due to off target effects of gefitinib on other kinases important for the survival of the CLL cells.

Despite clear functional activation of ZAP-70 mediated signalling, we show that ZAP-70 is not essential for survival or regulation of TCR related pathways in TCL/NKTL. It is possible

that this signal may instead be essential in mediating other NK-cell related functions. A recent report highlighted a role for ZAP-70 in reducing the cytotoxic activity of KHYG cells [38]. Here, depletion of ZAP-70 led to the downregulation of Granzyme B expression in KHYG which corresponded with reduced KHYG mediated cytotoxic activity against U266 cells. ZAP-70 may therefore play a more significant role in regulating the cytotoxic activity of NK cells, a question which requires evaluation in future studies.

The lack of differential gene expression and the absence of any significant enrichment of gene sets in 2 out of the 3 cell lines supports the conclusion that ZAP-70 is largely redundant in PTCL/NKTL. In both SNK1 and NKYS cell lines, except for the clear downregulation of ZAP70 expression, there was no other gene significantly dysregulated by the depletion of ZAP-70. Put together with the C2 gene set analysis, this gives a strong indication that the loss of ZAP70, particularly in the TCR signalling pathway can be compensated for by other factors. While the Hallmark gene set analysis identified a significant negative enrichment for E2F and MYC targets upon ZAP-70 depletion, this was not corroborated at the mRNA level via qRT-PCR. This is in keeping with the fact that there is no change in the cell survival and proliferative capacity seen in both ZAP-70 OE and KD cells.

The finding that cell cycle checkpoint gene expression was downregulated on ZAP-70 knockdown generated the hypothesis that overexpression of ZAP-70 may increase sensitivity to DNA damaging agents. Forced OE of ZAP-70 in the NK-YS cell line did not however lead to increased sensitivity to cytotoxic therapy. The initial observation that NK cells downregulate ZAP-70 in response to DNA damage was made in normal NK cells from healthy donors [26]. It is possible that malignant NK cells do not behave in the same manner, which may explain our findings. These findings also need to be validated in other NKTL and TCL cell lines as well as primary tumour samples.

## Conclusions and future directions

We conclude that ZAP-70 does not play a significant role in the survival and proliferation of T and NK cell lymphoma cell lines. Hence ZAP-70 is unlikely to be a suitable therapeutic target in these malignancies. Further studies on primary patient samples are required to verify our findings. Further work is also required to better understand the role of the T-cell signalling machinery downstream of ZAP-70 in PTCL and NKTL.

## Supporting information

**S1 Fig. Addition of OKT3 stimulates an increase in ZAP70 phosphorylation.** Activation of the TCR signaling pathway was stimulated by the addition of OKT3 10μg/mL over 4h and 18h. Immunoblot analyses was performed on these cell lines and the following proteins detected—phosphorylated and total ZAP70 as well as target substrates of ZAP-70, phosphorylated and total [LAT, SLP76, and p38 MAPK].
(PDF)

**S2 Fig. E2F and MYC gene targets, PRDX4, CMYC and ATF5 are not significantly downregulated upon ZAP70 knockdown.** (A) KHYG cells were treated with siNT or siZAP70 and then the effect on the mRNA expression levels of PRDX4, CMYC and ATF5 measured via qRT-PCR. (B) NKYS EV and ZAP70 overexpressing stable cell lines were analysed for levels of mRNA expression of PRDX4, CMYC and ATF5 measured via qRT-PCR.
(PDF)

**S3 Fig. Depletion of ZAP-70 in NKTL cell lines do not confer any sensitivity to Gefitinib treatment.** KHYG and HANK-1 cell lines which have been electroporated with either siNT or

siZAP70 were evaluated by a cell viability assay, Cell-Titer Glo (Promega) after 48h treatment with different doses of Gefitinib.
(PDF)

**S1 Raw images.**
(PDF)

## Author Contributions

**Conceptualization:** Sanjay de Mel, Nurulhuda Mustafa, Viknesvaran Selvarajan, Siok Bian Ng, Anand D. Jeyasekharan.

**Data curation:** Nurulhuda Mustafa, Muhammad Irfan Azaman, Zubaida Talal Alnaseri.

**Formal analysis:** Nurulhuda Mustafa, Muhammad Irfan Azaman, Zubaida Talal Alnaseri.

**Investigation:** Sanjay de Mel, Nurulhuda Mustafa, Muhammad Irfan Azaman, Patrick William Jaynes, Shruthi Venguidessane, Hoang Mai Phuong, Zubaida Talal Alnaseri, The Phyu, Joanna Wardyn, Ying Li, Omer An, Henry Yang, Siok Bian Ng, Anand D. Jeyasekharan.

**Methodology:** Sanjay de Mel, Nurulhuda Mustafa, Viknesvaran Selvarajan, Muhammad Irfan Azaman, Patrick William Jaynes, Shruthi Venguidessane, Hoang Mai Phuong, Zubaida Talal Alnaseri, The Phyu, Joanna Wardyn, Ying Li, Omer An, Henry Yang, Siok Bian Ng, Anand D. Jeyasekharan.

**Writing – original draft:** Sanjay de Mel, Nurulhuda Mustafa, Louis-Pierre Girard.

**Writing – review & editing:** Sanjay de Mel, Nurulhuda Mustafa, Viknesvaran Selvarajan, Muhammad Irfan Azaman, Patrick William Jaynes, Shruthi Venguidessane, Hoang Mai Phuong, Zubaida Talal Alnaseri, The Phyu, Louis-Pierre Girard, Wee Joo Chng, Joanna Wardyn, Ying Li, Omer An, Henry Yang, Siok Bian Ng, Anand D. Jeyasekharan.

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
