## [Decision Letter · Decision Letter 0]

7 Jun 2021

PONE-D-21-14507

T and NK cell Lymphoma cell lines do not rely on ZAP-70 in T-cell Receptor Signalling for Survival

PLOS ONE

Dear Dr. Sanjay de Mel,

Thank you for submitting your manuscript to PLOS ONE. After careful consideration, we feel that it has merit but does not fully meet PLOS ONE’s publication criteria as it currently stands. Therefore, we invite you to submit a revised version of the manuscript that addresses the points raised during the review process.

Specifically, the authors need to address evidence that reduced and over expression of ZAP-70 affects T cell receptor signaling and cell death. A time course following T cell receptor activation should be shown. Finally RNA-seq analysis needs to be further explained.

We look forward to receiving your revised manuscript.

Kind regards,

Spencer B. Gibson

Academic Editor

PLOS ONE

Journal Requirements:

2. Please include your tables as part of your main manuscript and remove the individual files. Please note that supplementary tables should remain uploaded as separate "supporting information" files.

Reviewers' comments:

Reviewer's Responses to Questions

**Comments to the Author**

1. Is the manuscript technically sound, and do the data support the conclusions?

Reviewer #1: No

Reviewer #2: Yes

2. Has the statistical analysis been performed appropriately and rigorously? 

Reviewer #1: Yes

Reviewer #2: Yes

3. Have the authors made all data underlying the findings in their manuscript fully available?

Reviewer #1: Yes

Reviewer #2: Yes

4. Is the manuscript presented in an intelligible fashion and written in standard English?

Reviewer #1: Yes

Reviewer #2: Yes

5. Review Comments to the Author

Reviewer #1: The manuscript entitled “T and NK cell lymphoma cells do not rely on ZAP-70 in T cell Receptor signaling for survival” for consideration for publication. The author knocked out ZAP-70 using siRNAand showed no change in apoptosis. Over expression of ZAP-70 failed to sensitize cells to etoposide induced cell death. This study is not sufficient to understand the role of ZAP-70 in T and NK cells. The rationale is described below.

1. There is no experimental evidence that reduced and over expression of ZAP-70 affects T cell receptor signaling. This could affect the interpretation of the results.

2. The use of siRNA showed reduced ZAP-70 expression failed to show a time course of knockdown and level of apoptosis.

3. Need to induce apoptosis in these cells to determine if the ability of these cells to undergo cell death.

4. For the gefitinib experiment, knockdown of ZAP70 and the effects of gefitinib treatment should be investigated.

5. RNAsequ shows affects in cell cycle (Myc or E2F targets) but nucleofector is very stressful on cells. This could alter cellular functions. Should be validated by over expression of ZAP-70. Should be evaluated following TCR signaling as well.

6. No functional validation of the results in RNAsequ.

Reviewer #2: The authors have investigated the role of Zap-70 in T-cell receptor signaling and survival thereof in T- and NK-cell lymphoma cell lines. Though highly expressed, down regulation of Zap-70 did not affect cell viability or apoptosis in the investigated cell lines. The down stream signaling that was affected on Zap-70 knockdown studied by RNA-seq analysis needs to be further explained. Potential hypothesis for pathways that may be causing Zap-70 independent TCR signaling can also be better explained in discussion. Immune dysregulation and whether PD1/PDL1 expression that was altered on Zap-70 manipulation in these cell lines if analyzed should be reported.

6. PLOS authors have the option to publish the peer review history of their article (what does this mean?). If published, this will include your full peer review and any attached files.

Reviewer #1: No

Reviewer #2: **Yes: **Praveen Ramakrishnan Geethakumari, MD

---

## [Author Response · Author response to Decision Letter 0]

3 Nov 2021

We are grateful to the reviewers and editorial team for their suggestions. We have made every attempt to address the points raised by the reviewers and modified the manuscript accordingly. We have included a detailed point by point reply to the reviewers as part of our submission entitled " response to reviewers".

---

## [Decision Letter · Decision Letter 1]

3 Dec 2021

T and NK cell Lymphoma cell lines are not reliant on ZAP-70  for Survival

PONE-D-21-14507R1

Dear Dr. de Mel,

We’re pleased to inform you that your manuscript has been judged scientifically suitable for publication and will be formally accepted for publication once it meets all outstanding technical requirements.

Kind regards,

Spencer B. Gibson

Academic Editor

PLOS ONE

Additional Editor Comments (optional):

Reviewers' comments:

Reviewer's Responses to Questions

**Comments to the Author**

1. If the authors have adequately addressed your comments raised in a previous round of review and you feel that this manuscript is now acceptable for publication, you may indicate that here to bypass the “Comments to the Author” section, enter your conflict of interest statement in the “Confidential to Editor” section, and submit your "Accept" recommendation.

Reviewer #1: All comments have been addressed

2. Is the manuscript technically sound, and do the data support the conclusions?

Reviewer #1: Yes

3. Has the statistical analysis been performed appropriately and rigorously? 

Reviewer #1: Yes

4. Have the authors made all data underlying the findings in their manuscript fully available?

Reviewer #1: Yes

5. Is the manuscript presented in an intelligible fashion and written in standard English?

Reviewer #1: Yes

6. Review Comments to the Author

Reviewer #1: Authors addresed all issues. the authors added new data and commented on specific questions. The manuscript is now acceptable for publication.

7. PLOS authors have the option to publish the peer review history of their article (what does this mean?). If published, this will include your full peer review and any attached files.

Reviewer #1: **Yes: **Spencer Gibson

---

## [Editor Report · Acceptance letter]

14 Jan 2022

PONE-D-21-14507R1 

T and NK cell Lymphoma cell lines do not rely on ZAP-70 for Survival 

Dear Dr. de Mel:

I'm pleased to inform you that your manuscript has been deemed suitable for publication in PLOS ONE. Congratulations! Your manuscript is now with our production department. 

Kind regards, 

on behalf of

Dr. Spencer B. Gibson 

Academic Editor

PLOS ONE